# Heart Rate Variability during Online Video Game Playing in Habitual Gamers: Effects of Internet Addiction Scale, Ranking Score and Gaming Performance

**DOI:** 10.3390/brainsci14010029

**Published:** 2023-12-26

**Authors:** Kehong Long, Xuzhe Zhang, Ningxin Wang, Hao Lei

**Affiliations:** 1State Key Laboratory of Magnetic Resonance and Atomic and Molecular Physics, Innovation Academy for Precision Measurement Science and Technology, Chinese Academy of Sciences, Wuhan 430071, China; kehonglong@126.com (K.L.); zhangxuzhe18@mails.ucas.ac.cn (X.Z.); ningxinhn@163.com (N.W.); 2University of Chinese Academy of Sciences, Beijing 100190, China; 3Wuhan National Laboratory for Optoelectronics, Huazhong University of Science and Technology, Wuhan 430074, China

**Keywords:** autonomic nervous system, heart rate variability, internet gaming disorder, multiplayer online battle arena (MOBA) game

## Abstract

Previous studies have demonstrated that individuals with internet gaming disorder (IGD) display abnormal autonomic activities at rest and during gameplay. Here, we examined whether and how in-game autonomic activity is modulated by human characteristics and behavioral performance of the player. We measured heart rate variability (HRV) in 42 male university student habitual gamers (HGs) when they played a round of League of Legends game online. Short-term HRV indices measured in early, middle and late phases of the game were compared between the players at high risk of developing IGD and those at low risk, as assessed by the revised Chen Internet addiction scale (CIAS-R). Multiple linear regression (MLR) was used to identify significant predictors of HRV measured over the whole gameplay period (WG), among CIAS-R, ranking score, hours of weekly playing and selected in-game performance parameters. The high-risk players showed a significantly higher low-frequency power/high-frequency power ratio (LF/HF) relative to the low-risk players, regardless of game phase. MLR analysis revealed that LF/HF measured in WG was predicted by, and only by, CIAS-R. The HRV indicators of sympathetic activity were found to be predicted only by the number of slain in WG (N_Slain_), and the indicators of parasympathetic activity were predicted by both CIAS-R and N_Slain_. Collectively, the results demonstrated that risk of developing IGD is associated with dysregulated autonomic balance during gameplay, and in-game autonomic activities are modulated by complex interactions among personal attributes and in-game behavioral performance of the player, as well as situational factors embedded in game mechanics.

## 1. Introduction

Internet gaming disorder (IGD) has been officially listed as a mental disorder in the 11th revision of the International Classification of Diseases (ICD-11) published by the World Health Organization [1]. IGD is characterized by a pattern of persistent or recurrent Internet-based or offline gaming behavior that is manifested by impaired control over the behavior, giving increasing priority to gaming to the extent that it takes precedence over other life interests and daily activities; and continuing or escalating gaming despite its negative consequences [1]. Numerous neuroimaging studies have demonstrated that IGD is associated with various abnormalities in the central nervous system (CNS), manifesting as structural plasticity [2,3,4] and altered resting-state functional connectivity in specific brain areas/networks [5,6,7], as well as aberrant functional activations in tasks involving executive control, impulse inhibition and working memory [8,9,10].

Abnormal autonomic activities have also been reported in IGD and problematic internet use (PIU) [11,12]. PIU has currently no official diagnostic criteria, but can be defined as use of Internet that created psychological, social, school and/or work difficulties in a person’s life [13]. The autonomic nervous system (ANS) is coupled to the CNS anatomically and functionally [14]. The ANS receives top-down regulation from the CNS [15,16], and provides feedback through the ascending pathways [17,18]. The ANS plays an important role in regulating the majority of the body’s internal processes, allowing adaptive responses to internal and external stressors and guaranteeing the body’s homeostatic milieu [14]. The dynamic interactions between the CNS and ANS modulate perception, cognition, emotion generation/regulation, memory consolidation and even the sense of self [16,19,20,21,22].

Heart rate variability (HRV) is probably the most widely used measurement to assess ANS activity by quantifying the time interval variations between adjacent heart beats [23,24,25]. Sympathetic/parasympathetic tones and prevalence are represented by different HRV indices computed in the time, frequency and non-linear domains [26]. HRV measurements have been used in laboratory [27,28] and real-world tasks [29,30] to assess ANS activity perturbations associated with cognitive load [31], mental stress [32], emotional arousal [33], executive control [34] and reactivity to reward [35]. Moreover, it has been proposed that HRV can serve as an auxiliary diagnostic tool for certain mental diseases [27,36].

Compared to none-gamer or occasional/habitual player controls without IGD, players with IGD were reported to have higher heart rates (HRs), lower standard deviation of inter-beat intervals (SDNN), lower root mean square differences between successive inter-beat intervals (RMSSD), as well as lower high-frequency power percentage and higher low-frequency power percentage at rest, indicating a higher sympathetic tone and/or lower parasympathetic/vagal tone in this population [37,38,39,40,41]. IGD-related HRV abnormalities at rest were found to correlate with distressed personality total scores [38] and carving ratings [39], and thought to be linked to defective inhibitory control and reduced autonomic flexibility [42,43]. A recent meta-analysis concluded that individuals with PIU have significantly different resting state parasympathetic activity from healthy controls [11].

Consistent with the observations in those with alcohol addiction and gambling disorder [44,45], individuals with IGD also appeared to have higher HRV reactivity to game cues and during gameplay when compared to individuals without IGD. IGD individuals showed larger changes in low-frequency power/high-frequency power ratio (LF/HF) upon exposure to visual game cues [46]. A significant decrease in high-frequency power (HF) during gameplay was observed in IGD individuals, but not in healthy controls [47]. When playing a multiplayer online battle arena (MOBA) game, League of Legends (LOL), only individuals with IGD showed significant in-game HF decrease, the amplitude of which correlated negatively with IGD severity assessed by internet addiction test score [48,49].

Game content and context are known to affect in-game HRV as well. Playing violent games induced greater changes in HRV compared to playing non-violent games [50,51]. Playing a fighting game was associated with lower in-game RMSSD than playing puzzle games, and the players receiving threat appraisal instructions before gameplay showed significantly lower RMSSD than those received challenge appraisal instructions [52]. Evidence also suggests that in-game HRV abnormalities observed in IGD individuals were situational, with different manifestations in different periods/phases of the game. For instance, some authors reported that IGD individuals had significant HF decreases in the early phase of the game when gameplay was still relatively easy and not complicated by real-time state of gaming [48,53], while others showed that HF in IGD individuals did not change in the early game phase, but instead decreased significantly when high attention was required, as well as in the last minutes of gameplay when game ending was approaching [49]. Like in-game CNS activities, in-game ANS activities/abnormalities observed in these studies might be attributed to dynamic and complicated interactions between human factors of the player and mechanics of the game being played [54,55].

Few previous studies have measured in-game HRV in habitual video gamers who are not formally diagnosed with IGD, or examined the factors influencing the in-game HRV in this population. In this study, real-time HR in habitual gamers (HGs) was recorded continuously with a wearable electroencephalogram (EEG) device when playing a round of LOL game under ergonomic and naturalistic conditions. The HR data were analyzed in two different ways. In the first analysis, short-term HRV in three 5-min time windows representing early (EG), middle (MG) and late (LG) phases of gameplay were computed and compared between the individuals at high risk of developing IGD and those at low risk. The aim of the first analysis is to test whether and how in-game HRV is affected by the risk of developing IGD. In the second analysis, multiple linear regression (MLR) was used to identify significant predictors of in-game HRV measured over the whole gameplay period (WG), aiming to test whether in-game HRV and ANS activities are affected by in-game behavioral performance and human characteristics of the player.

## 2. Materials and Methods

### 2.1. Participants

Forty-two right-handed male university students having at least 2 years of experience on LOL playing and, at the same time, having played more than 1000 LOL matches in the past were recruited. Written informed consent and demographic information (Table 1) were obtained from each participant before starting the experiments. The risk of developing IGD was assessed using the revised Chen Internet addiction scale (CIAS-R) [56]. The skill level of the player in LOL was assessed using the ranking score provided by the official ranking system at the end of last LOL season (i.e., Season 2018). Self-reported hours of weekly LOL playing (HoWP) was used to assess the frequency of LOL playing in the past year. None of the participants had any neurological or psychiatric history, medical history affecting their ANS or sleep deprivation when they participated the study. All participants had normal or corrected normal vision, and were reported to have not used any neurological or psychiatric drugs in the 24 h before the experiment. The research was approved by Human Subjects Institutional Review Board of Wuhan Institute of Physics and Mathematics, Chinese Academy of Science (Protocol Number: WIPM20190303).

### 2.2. Experimental Procedures

Each participant was asked to play a ranking-mode LOL match on a Dell computer (OptiPlex 3020SSF) with a 23.8-inch screen (DELL E2417H) in a sound-proof room, together with 9 anonymous online players selected automatically by the built-in matchmaking algorithm in the game. A chest strap equipped with a Polar H10 ECG sensor (Polar Electro Oy, Kempele, Finland) was used for HR recording at a sampling rate of 130 Hz [57]. The HR data were recorded continuously for 1 min before the game onset and throughout the entire period of gameplay. No bystander was present in the room except the experimenters. Video data of the games were recorded at 15 frames/s using a built-in function provided by LOL.

### 2.3. HRV Measurement

The HR data were extracted from the Polar Best app (versions 2.3.1) and transformed into R-R intervals. The Kubios HRV Standard freeware (versions 3.3) was used to calculate HRV indices from the R-R interval time series [58]. Square root of Baevsky’s stress index (STRESS) [59], three time domain indices (i.e., Mean HR, SDNN and RMSSD) and one nonlinear index (i.e., sample entropy (SampEn)) were calculated, as well as four frequency domain indices calculated with the autoregressive model, including low-frequency power (LF), HF, total power (TP) and LF/HF.

### 2.4. LOL Game

LOL is one of the most popular MOBA video games worldwide [60]. It is known that playing MOBA games is associated with more time spent on video gaming and a higher endorsement of IGD symptoms [61]. In each LOL game, 10 players summoned are divided into 2 ad hoc teams. Each player selects and controls his/her own avatar (or champion) to fight with the champions and no-player-controlled characters from the opponent team in a virtual game arena. To win the game, one team needs to destroy the base (or nexus) of the other team while preventing its own base being destroyed. A typical LOL game lasts 20–40 min and ends when the base of one team is destroyed or one team surrenders voluntarily.

There is a seven-tier (i.e., Bronze, Silver, Gold, Platinum, Diamond, Master and Challenger) ranking system in LOL (Version 8.22), and within each tier there are five divisions, with five being the lowest in that tier and one being the highest. Game rank of the participants in this study ranged from Bronze to Master, with 5 in Bronze, 18 in Silver, 2 in Platinum, 16 in Diamond and 1 in Master. For statistical analyses, the categorical game ranks of the players were transformed into real number ranking scores using a linear coding system, with 1 representing tier Bronze and 6 representing tier Master. An increment of 0.2 was used to code each within-tier divisional promotion. For example, Division 5 in tier Sliver was coded as 2.0, and Division 3 in tier Diamond as 5.4.

A typical LOL game can be divided roughly into EG, MG and LG phases, based on the interim goals and the gaming strategy the player would most likely adopt. Since there are no clear markers that can be used to tell the game phases apart, we defined them operationally. The 5 min immediately after game onset was taken as the EG phase, the 5 min around the midpoint of each game as the MG phase, and the last 5 min of the game as the LG phase. In-game behavioral parameters used for data analysis, including game length, number of slay (N_slay_) and number of slain (N_slain_), were read directly from the game log.

### 2.5. Data Analysis

The HRV data were analyzed using two different methods. In the first analysis, the participants were divided into two groups according to their CIAS-R. The participants with a CIAS-R higher than 63 were considered to have a high risk of becoming addicted [56] and therefore assigned into the High-Risk group (n = 20), while those with a CIAS-R lower than 63 were assigned into the Low-Risk group (n = 22) (Table 1). An independent sample t-test or Mann–Whitney U test was used for statistical analysis of the inter-group differences in demographic and in-game behavioral parameters. Repeated measures analysis of covariance (RM-ANCOVA) was used to compare HRV measured in different game phases, with group as the between-subjects factor (High Risk vs. Low Risk), game phase as the within-subjects factor. All demographic parameters listed in Table 1, except CIAS-R, was used as covariates. Independent-sample and paired t-tests were used for post hoc between-subject and within-subject comparisons, respectively. False discovery rate (FDR) correction was applied to control type-I errors associated with multiple post hoc comparisons.

In the second analysis, MLR was used to identify significant predictors for in-game HRV measured over WG (27 ± 5 min). All demographic and behavioral parameters listed in Table 1 were included in the regression model, except education and game length. Education had a high correlation with age (r = 0.794, *p* < 0.001), and game length showed significant correlations with both N_slay_ (r = 0.327, *p* = 0.034) and N_slain_ (r = 0.418, *p* = 0.008). Collinearity among the included parameters was assessed using the variation inflation factor (VIF). In cases where more than one significant predictor was identified, mediation and moderation analyses were used to examine the relationships between/among the significant predictors, controlling for the effects of all the other variables included in the MLR model.

Normality of all data was verified with the Shapiro–Wilk tests. A *p* ≤ 0.05, corrected if applicable, was considered to be statistically significant. All statistical analyses were performed with the software package SPSS 20.0 (PC version).

## 3. Results

A total of 42 subjects were included in this study. Table 1 lists the demographic and in-game behavioral parameters of these subjects.

### 3.1. Analysis 1: Game Phase-Dependent Changes of In-Game HRV

The High-Risk group and Low-Risk group showed significant inter-group difference in CIAS-R (t = −10.865, *p* < 0.001), but not in any other demographic parameters (Table 1). Neither did The two groups show any statistically significant inter-group differences in game length, N_slay_ or N_slain_ (Table 1).

Analyzing the data with RM-ANCOVA yielded statistically significant main effect of group for LF/HF (F(1,37) = 4.069, *p* = 0.050, η^2^ = 0.099, Figure 1a), but not for LF (F(1,37) = 0.303, *p* = 0.586, η^2^ = 0.008, Figure 1b), HF (F(1,37) = 2.519, *p* = 0.121, η^2^ = 0.064, Figure 1c) or any other HRV indices. A statistically significant main effect of game phase × group interaction was observed for STRESS (F(1,37) = 3.285, *p* = 0.043, η^2^ = 0.082). STRESS tended to be higher in EG than in MG (*p* = 0.073, FDR corrected) in the High-Risk group, while the Low-Risk Group did not show this trend (Figure 1d). No HRV index showed statistically significant main effects of game phase or game phase × covariant interactions, suggesting that statistically in-game HRV was not game phase-dependent.

### 3.2. Analysis 2: Multiple Linear Regression of HRV Measured over WG

The results of enter MLR analysis of HRV measured over WG are listed in Table 2. The prediction models for HF (R^2^ = 0.392, F(6,35) = 3.761, *p* = 0.005), RMSSD (R^2^ = 0.390, F(6,35) = 3.731, *p* = 0.006) and STRESS (R^2^ = 0.303, F(6,35) = 2.534, *p* = 0.038) were statistically significant. HF was predicted by CIAS-R (β = −0.279, *p* = 0.040), ranking score (β = −0.472, *p* = 0.009) and N_slain_ (β = −0.532, *p* = 0.003). RMSSD was predicted by CIAS-R (β = −0.265, *p* = 0.047) and N_slain_ (β = −0.560, *p* = 0.002). STRESS was predicted by, and only by, N_slain_ (β = 0.519, *p* = 0.006). Without having statistically significant prediction models, LF/HF (β = 0.430, *p* = 0.008) was predicted by, and only by, CIAS-R. SDNN (β = −0.545, *p* = 0.005), LF (β = −0.517, *p* = 0.009) and TP (β = −0.562, *p* = 0.004) was predicted by, and only by, N_slain_. All parameters included in the regression analysis had a VIF < 5, indicating that there was no significant collinearity among them in the regression analyses [62]. Figure 2 plots LF/HF, LF, HF and STRESS against CIAS-R and N_slain_, as well as the linear fits to the data.

Neither the HF vs. CIAS-R correlation nor the RMSSD vs. CIAS-R correlation was moderated/mediated by N_Slain_, nor was the HF vs. CIAS-R correlation moderated/mediated by ranking score. However, the correlation between HF and ranking score was found to be partially mediated by N_slain_, and vice versa. The mediating effects of N_slain_ (total effect = −1.950, *p* = 0.407; direct effect = −6.908, *p* = 0.010; indirect effect = 4.958, *p* < 0.001) and ranking score (total effect = −2.173, *p* = 0.056; direct effect = −4.175, *p* = 0.002; indirect effect = 2.002, *p* < 0.001) on each other manifested as a masking effect, given the opposite signs of the direct and indirect effects [63]. Assessing pairwise correlations among the parameters entered into MLR yielded a negative correlation between N_slain_ and ranking score (r = −0.507, *p* = 0.001, Spearman’s correlation), which was also the only pairwise correlation that was statistically significant.

## 4. Discussion

The current study aimed to characterize in-game HRV in male university student HGs when they played a round of an LOL game, and explore how the in-game HRV was modulated by human characteristics, especially CIAS-R, and in-game performance of the player. When dividing the players into the High-Risk and Low-Risk groups based on CIAS-R, it was shown that the High-Risk group had significantly higher LF/HF during gameplay, regardless of the game phase. The players in the High-Risk group, but not in the Low-Risk group, also tended to have higher STRESS in EG than in MG. Analyzing the data from all players together, a significant predictor for in-game HRV indices measured over WG was identified with enter MLR. LF/HF was found to be predicted by, and only by, CIAS-R. In comparison, HRV indices commonly used as indicators of sympathetic activity, such as STRESS, SDNN and TP, were predicted by, and only by, N_Slain_. HF and RMSSD, two HRV indices commonly used as indicators of parasympathetic activity, were predicted by both CIAS-R and N_slain_, and additionally HF was also predicted by ranking score. HoWP had little influence on in-game HRV.

The results of the two analyses demonstrated, convergently, that the players at high risk of developing IGD had higher LF/HF during gameplay (Figure 1a and Table 2). HF/LF is an HRV index frequently used to assess the balance between the sympathetic and parasympathetic branches of the ANS [59,64,65]. Previous research has reported that individuals with IGD have higher LF/HF at rest compared to healthy controls [37,47]. Interestingly, individuals with alcohol and methamphetamine dependence are also known to have higher LF/HF at rest, presumably reflecting a shifted autonomic balance towards a sympathetically dominated state [66] or cardiotoxic/neurotoxic autonomic dysbalance [67]. The results obtained in this study are consistent with and expand upon these previous findings, by showing that the players at high risk of developing IGD (i.e., CIAS-R > 63), but yet to be officially diagnosed with IGD, had elevated LF/HF during online video game playing. Moreover, it was demonstrated that LF/HF was higher in the players at high risk of developing IGD in all three game phases, suggesting that this feature may not be less related to the task load and cognitive demand of gameplay, but rather a personal attribute of the player, which would manifest both at rest and during gameplay.

Elevated in-game LF/HF in the players at high risk of developing IGD appeared to be driven mainly by reduced HF. In Analysis 1, HF appeared to be lower in the High-Risk group than in the Low-Risk group (Figure 1c), although the difference did not reach statistical significance. In Analysis 2, CIAS-R was identified to be a significant predictor of HF, but not of LF (Table 2). Like RMSSD, HF is a well-established indicator of parasympathetic (vagal) tone [64,68]. In comparison, the interpretation of LF is more complicated. It is argued by some authors that LF is related to both sympathetic and parasympathetic activities [65,69]. Both HF and RMSSD were negatively predicted by CIAS-R, suggesting that in-game parasympathetic activity was lower in the players at high risk of developing IGD. This observation is consistent with previous reports showing that only subjects with IGD, but not healthy controls, had significant HF decreases during gameplay compared to pre-game baseline [48,49].

STRESS is sensitive to amplification of sympathetic tone and considered an HRV index reflecting the activity of sympathetic regulation gears [59]. In Analysis 1, the High-Risk group showed higher STRESS in EG than in MG (Figure 1d). This observation is in line with previous reports showing that individuals with IGD have significant decreases in HF during the first 5 min of LOL playing [48,53]. On the other hand, it seemingly contradicts the finding of Hong et al., who reported no significant HF changes in the first 5 min in the IGD subjects [49]. The mechanisms underlying game phase-related sympathetic/parasympathetic activation in the players at high risk of developing IGD remain to be elucidated. Increased reactivity to game cues, such as anticipation and anxiety, might have given rise to elevated STRESS in EG in the players at high risk of developing IGD. Previous studies have demonstrated that exposure to visual game cues is sufficient to induce significant sympathetic activations in individuals with IGD, manifesting as increased respiratory rate and decreased standard deviation of HR [70]. Upon exposure to game cues, individuals with IGD also exhibited higher activations in distributed brain regions in the fronto–striatal–limbic networks, which are known to be involved in planning, motivational encoding and reward processing [71,72,73]. Attenuated impulse control might have contributed to enhanced reactivity of the CNS and ANS to game cues in these individuals [74,75,76].

A number of HRV indices were found to be predicted significantly by N_slain_, including both the indicators of parasympathetic (i.e., HF and RMSSD) activity and those of sympathetic (i.e., STRESS, SDNN and TP) activity. STRESS increases with sympathetic activation [59], and TP and SDNN decrease with sympathetic activation [77,78,79]. Collectively, the results listed in Table 2 suggest that in-game sympathetic activity increases with N_Slain_, while in-game parasympathetic activity decrease with N_Slain_. Slain is generally considered as a negative event in LOL. The player showing a count of slain that is above the team average is often considered an easy pick and often targeted at by the opponent team deliberately. At the same time, he or she may face destructive criticism from the teammates and a risk of being reported as a feeder or griefer. N_slain_ represents the total number of times a player is slain in WG. Therefore, having a high N_slain_, for most of the time, would mean the player has had poor performance and unpleasant experience in the game.

Numerous studies have shown that task-related changes in HRV are modulated by physical/cognitive demands, subjectively perceived difficulty, mental stress and task performance [31,80,81,82]. Consistently, these studies show that tasks with higher difficulty and heavier cognitive demand are in general associated with higher level of mental stress, augmented sympathetic activations and weakened parasympathetic activities [31,83,84]. Our observations concerning the prediction of in-game HRV by N_Slain_ are consistent with these previous findings, supporting the notion that the poorer the player’s performance during gameplay, the more stressed he/she may feel, consequently leading to enhanced sympathetic activities and reduced parasympathetic activities. While the in-game sympathetic activity was mainly determined how well the player performed during gameplay, the in-game parasympathetic (vagal) activity is modulated by both behavioral performance and personal attributes of the player, as evidenced by the prediction of RMSSD and HF by both N_Slain_ and CIAS-R. These results suggest that the sympathetic and parasympathetic branches are likely regulated through dissociable psychophysiological pathways during online video game playing, with the parasympathetic branch more closely related to the propensity of developing IGD.

Additionally, HF was negatively predicted by ranking score (Table 2). It should be noted that N_Slain_ correlated negatively with ranking score as well, and the correlation between HF and ranking score was found to be partially mediated by N_Slain_, and vice versa. Both the direct effects and indirect effects of ranking score and N_Slain_ on HF were statistically significant, albeit in opposite signs. This means that ranking score and N_Slain_ can modulate HF significantly and independently, despite that the effects of the two tend to mask each other. In other words, the high-ranking players would show an in-game HF lower than that observed in the low-ranking players, when they have the same N_Slain_; and the players with a high N_Slain_ would have an in-game HF lower than that observed in those with a low N_Slain_, when they have the same ranking score.

The negative correlation between ranking score and in-game HF could have arisen from the game mechanics of LOL. The matchmaking algorithm of LOL summons only those with comparable gaming records and skill levels to play together in a game, and in principle this practice customizes the game difficulty for each participant. However, the gameplay intensity and cognitive load in absolute terms may, still, vary among the games played by the players at different ranks. In an MOBA game, the collective actions of all players influence how the game unfolds and how competitive it is. It therefore may be true that the high-ranking players need to be more immersed and use higher and more complicated cognitive exertions, such as extra vigilance, divided attention and multi-tasking, to achieve a favorable outcome, as the margins for lapses and errors are narrower for them than for those at lower ranks. It is known that increases in immersion (i.e., flow), cognitive demand and “adrenaline-rush” experience during video game playing give rise to the so-called “flow” state, which is characterized by parasympathetic withdrawal-associated arousal and a decrease in HF [85,86,87,88]. Consistent with our observation, previous studies have demonstrated that expert neurointerventionalists have lower HF than non-experts when performing simulated neuroangiographies with the same workload [89], and familiarized chess players have lower HF than unfamiliarized chess players when playing chess in a strange computer environment [30]. It should be noted that there are a number of reports in the literature showing that experts tend to have higher HF than non-experts at rest [90,91] and the individuals having a higher HF at rest often display superior performance in cognitive tasks, such as N-back task and Stroop task [92,93]. High-performance chess players are shown to have higher RMSSD than low-performance chess players when playing chess at low- to medium-difficulty levels, but not at high-difficulty levels [31]. These results highlight that task-related changes in parasympathetic activity are most likely modulated by the complex interactions among the expertise, task difficulty and contingency of behavioral performance, rather than expertise per se.

There are several limitations that need to be addressed. Firstly, complementary measurements of physiological indices such as respiration rate, blood pressure and skin conductance are required to provide a more complete and comprehensive understanding on how in-game autonomic activities are affected by in-game behaviors and personal attributes of the player. For example, respiratory parameters are important determinants of respiratory sinus arrhythmia (RSA), which may affect LF and HF independent of changes in cardiac vagal tone. Decreased respiratory rate is usually accompanied with increased LF [94]. Therefore, the possible contribution of altered respiratory rhythm to the relatively higher LF/HF observed in the players at high risk of developing IGD cannot be excluded, given the reports showing that these players tended to have a lower respiration rate and/or hold breath more often during VGP [95]. Further studies are needed to clarify how changes of breath rhythm affect the measurement and interpretation of in-game HRV. Secondly, since baseline HRV was not measured in this study, it remained unclear to what extent in-game HRV was related to and influenced by resting state HRV. A, LOL game typically lasts 20–40 min. Our preliminary experiments showed that it is challenging for subjects to remain in a resting state for an extended period of time that is comparable to the actual gameplay (i.e., tens of minutes). Thirdly, the data were obtained from a relatively small and homogenous group of players who played a single game. Further research is required to investigate whether the observed results can be generalized to other player populations and games. Finally, the precision, accuracy and interpretation of HRV measurements are known to depend on the length of the time window during which the heart beats are recorded [79]. HRV measurements at various timescales may capture different aspects of in-game autonomic changes and their associations with in-game behaviors and psychophysiological states [96].

## 5. Conclusions

HRV measurements were used to detect changes in autonomic activities during online video game playing in male university student HGs. In-game LF/HF, an HRV indictor of autonomic balance, was found to be higher in the players at high risk of developing IGD, giving further support to the notion that this HRV index may potentially be used for IGD screening and/or diagnosis. Poor performance during gameplay was found to be linked to higher sympathetic activity and lower parasympathetic activity. The in-game performance also, in part, mediated the modulating effect of ranking score on HRV indicators of parasympathetic activity, including HF. Last but not least, the players at high risk of developing IGD tended to have higher STRESS in EG than MG, probably reflecting enhanced reactivity to game cues in this population. Together, the results of this study suggest that IGD-related autonomic dysregulation may have arisen from the complex interactions among personal attributes and in-game behavioral performance of the player, as well as factors embedded in the mechanics of the game being played.

## Figures and Tables

**Figure 1 brainsci-14-00029-f001:**
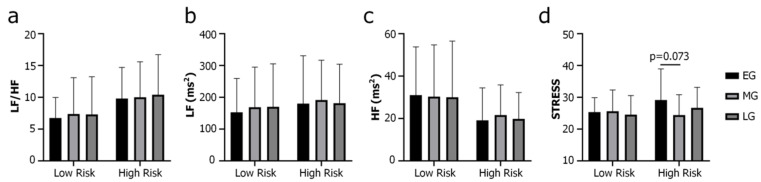
Short-term HRV measurements in early (EG), middle (MG) and late (LG) phases of the game, including LF/HF (**a**), LF (**b**), HF (**c**) and STRESS (**d**). LF/HF: low-frequency power/high-frequency power ratio, LF: low-frequency power, HF: high-frequency power, STRESS: square root of Baevsky’s stress index. *p* value displayed in (**d**) was false discovery rate corrected for multiple (n = 9) post hoc comparisons.

**Figure 2 brainsci-14-00029-f002:**
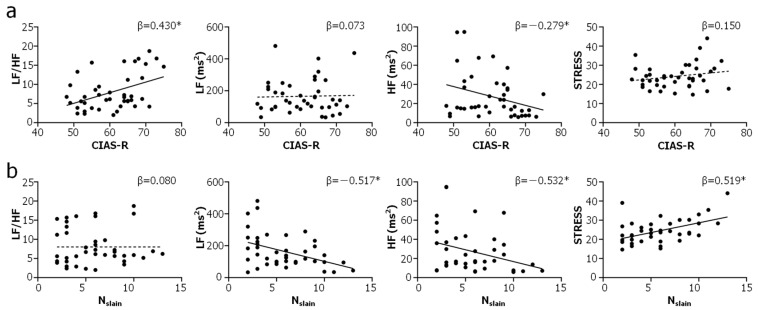
Plots of selected HRV indices against CIAS-R (**a**) and N_slain_ (**b**). LF/HF: low-frequency power/high-frequency power ratio, LF: low-frequency power, HF: high-frequency power, STRESS: square root of Baevsky’s stress index, β: standardized regression coefficient obtained from enter multiple linear regression. * *p* < 0.05. Solid and dashed lines: linear fits to the data, with a statistically significant β represented by a solid line, and an insignificant β represented by a dashed line.

**Table 1 brainsci-14-00029-t001:** Demographics and in-game behavioral parameters.

Parameters	High Risk (n = 20) M (SD)	Low Risk (n = 22) M (SD)	*p* Value
Age (years)	20.4 (2.4)	20.0 (1.9)	0.648
Education (years)	14.8 (2.1)	14.4 (1.7)	0.512
HoWP (hours)	13.4 (7.5)	11.3 (6.8)	0.348
Ranking score	3.5 (1.5)	3.7 (1.6)	0.736
CIAS-R	67.3 (3.4)	54.6 (4.1)	<0.001 *
Game length	26.8 (5.7)	27.4 (4.7)	0.685
N_slay_	7.3 (4.5)	5.3 (4.2)	0.140
N_slain_	6.4 (3.6)	5.2 (2.3)	0.218

M: mean, SD: standard deviation, HoWP: self-reported hours of weekly LOL playing, CIAS-R: revised Chen Internet Addiction Scale, N_slay_: number of slay, N_slain_: number of slain. *: *p* < 0.05.

**Table 2 brainsci-14-00029-t002:** Standardized regression coefficients (β) in the multiple linear regression analysis with the enter method.

Parameters	Age	CIAS-R	HoWP	Ranking Score	N_slay_	N_slain_
Mean HR	−0.184	0.117	0.094	−0.206	−0.180	−0.034
LF/HF	0.100	0.430 *	−0.034	0.238	0.070	0.080
LF	−0.129	0.073	0.012	−0.166	0.048	−0.517 *
TP	−0.151	0.024	−0.017	−0.231	−0.006	−0.562 *
SDNN	−0.160	−0.010	−0.021	−0.222	0.012	−0.545 *
STRESS ^#^	0.121	0.150	0.078	0.086	0.004	0.519 *
HF ^#^	−0.201	−0.279 *	0.008	−0.472 *	−0.114	−0.532 *
RMSSD ^#^	−0.166	−0.265 *	−0.071	−0.252	0.040	−0.560 *
SampEn	0.069	0.069	−0.198	−0.218	−0.139	−0.081

CIAS-R: revised Chen Internet Addiction Scale, HoWP: self-reported hours of weekly LOL playing, N_slay_: number of slay, N_slain_: number of slain. HR: heart rate, LF/HF: low-frequency power/high-frequency power ratio, LF: low-frequency power, TP: total power, SDNN: standard deviation of inter-beat intervals, STRESS: square root of Baevsky’s stress index, HF: high-frequency power, RMSSD: root mean square differences between successive inter-beat intervals, SampEn: sample entropy. ^#^: regression model being statistically significant, *: *p* < 0.05.

## Data Availability

The data presented in this study are available on request from the corresponding author. The data are not publicly available due to the fact that they constitute an excerpt of research in progress.

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
