# Peer review of "Heart Rate Variability during Online Video Game Playing in Habitual Gamers: Effects of Internet Addiction Scale, Ranking Score and Gaming Performance"

_brainsci, 2023, doi:10.3390/brainsci14010029_

Round 1
Reviewer 1 Report
Comments and Suggestions for Authors
1.The methodology has problems with the use of HRV indicators - the duration of the cardiorhythmogram recording and the lack of data on the respiratory rate.
Thus, the definition of low-frequency LF HRV indicators of less than 2 minutes (1 minute) is incorrect (Line 127). Only high-frequency HRV (HF) indicators will be correctly assessed on such ultra-short cardiorhythmogram recordings (less than 2 minutes). As a compromise, the authors should write about the average duration of the LF wave period (0.15–0.04 Hz, oscillation periods from 6.7 to 25 s). That is, there must be confidence that during a 1-minute recording of a cardiorhythmogram there were at least 2-3 periods of the LF wave. This is a serious limitation of the study (short recordings of cardiorhythmograms up to 2 minutes), which the authors need to reflect at the end of the article, in the Discussion section.
Respiratory frequency determines the peak frequency of the LF component of HRV in healthy people. If breathing is less than 9 respiratory cycles per minute, then the LF peak automatically shifts to the low frequency range and does not always correlate with the sympathetic activity of the autonomic regulation of heart rate. In this case, sympathetic activity will be reflected not by spectral, but by temporal indicators of HRV (for example, Mode Amplitude) or geometric indicators (for example, Baevsky’s stress index). When working with a computer for a long time, especially gamers, they hold their breath. What is the authors' opinion about Screen Apnea Syndrome? This is why LF/HF in gamers and/or people with high risk of Internet addiction may be higher than in the control group, although the Stress Index was not statistically different in groups with different risks of Internet addiction (Figure 1). Not only the lack of information on respiratory rate, but also the lack of consideration of respiratory rate when interpreting HRV indicators, especially in gamers, is a serious limitation of the study. The LF/HF indicator in this article cannot be a decisive (fundamental) indicator that adequately reflects the sympathetic influences on the heart rhythm.
2. Considering that the cardiorhythmogram was recorded during a non-stationary process, it is appropriate to consider the stress of cardiac activity using nonlinear methods for assessing HRV. The lack of nonlinear HRV measures in the context of this study is a major limitation of the study.
Author Response
1.The methodology has problems with the use of HRV indicators - the duration of the cardiorhythmogram recording and the lack of data on the respiratory rate.
Reply: Thank you for your helpful comments, we will provide detailed reply to these problems below.
Thus, the definition of low-frequency LF HRV indicators of less than 2 minutes (1 minute) is incorrect (Line 127). Only high-frequency HRV (HF) indicators will be correctly assessed on such ultra-short cardiorhythmogram recordings (less than 2 minutes). As a compromise, the authors should write about the average duration of the LF wave period (0.15–0.04 Hz, oscillation periods from 6.7 to 25 s). That is, there must be confidence that during a 1-minute recording of a cardiorhythmogram there were at least 2-3 periods of the LF wave. This is a serious limitation of the study (short recordings of cardiorhythmograms up to 2 minutes), which the authors need to reflect at the end of the article, in the Discussion section.
Reply: We quite agree with the opinion of the reviewer. But in this study, the cardiorhythmogram were recorded from 1 minute before the game onset to game over, and the averaged cardiorhythmogram recording time more than 20 minutes, rather than 1 minute. In data analysis, the short-term HRV indices were computed from 5 minutes-long cardiorhythmogram recording. Thus, both low-frequency (LF) and high-frequency (HF) indicators can be correctly assessed on this 5 minutes-long cardiorhythmogram recordings. Meanwhile, we discussed the impact of cardiorhythmogram recording time scale on the HRV measurements in the ‘Discussion’ section (Line 432) as well.
Respiratory frequency determines the peak frequency of the LF component of HRV in healthy people. If breathing is less than 9 respiratory cycles per minute, then the LF peak automatically shifts to the low frequency range and does not always correlate with the sympathetic activity of the autonomic regulation of heart rate. In this case, sympathetic activity will be reflected not by spectral, but by temporal indicators of HRV (for example, Mode Amplitude) or geometric indicators (for example, Baevsky’s stress index). When working with a computer for a long time, especially gamers, they hold their breath. What is the authors' opinion about Screen Apnea Syndrome? This is why LF/HF in gamers and/or people with high risk of Internet addiction may be higher than in the control group, although the Stress Index was not statistically different in groups with different risks of Internet addiction (Figure 1). Not only the lack of information on respiratory rate, but also the lack of consideration of respiratory rate when interpreting HRV indicators, especially in gamers, is a serious limitation of the study. The LF/HF indicator in this article cannot be a decisive (fundamental) indicator that adequately reflects the sympathetic influences on the heart rhythm.
Reply: We quite agree with the opinion of the reviewer. The lack of data on respiratory rate is the major limitation of this study. We have added and discussed this limitation at the end of the article, in the ‘Discussion’ section (Line 419). Respiratory parameters are important determinants of respiratory sinus arrhythmia (RSA), which may affect LF and HF independent of changes in cardiac vagal tone. Decreased respiratory rate is usually accompanied with increased LF. Therefore, the possible contribution of altered respiratory rhythm to the relatively higher LF/HF observed in the players at high risk of developing IGD cannot be excluded, given the reports showing that these players tended to have lower respiration rate and/or hold breath more often during video game playing.
2. Considering that the cardiorhythmogram was recorded during a non-stationary process, it is appropriate to consider the stress of cardiac activity using nonlinear methods for assessing HRV. The lack of nonlinear HRV measures in the context of this study is a major limitation of the study
Reply: We added a nonlinear HRV index (i.e., sample entropy) in the revised manuscript (Line 152 and Table 2).

Reviewer 2 Report
Comments and Suggestions for Authors
The authors presented an interesting study illustrating the correlates of an ANS evaluation performed through HRV in HG. The manuscript is modern and will raise readers interest, but there are few minor concerns that may increase the overall quality if addressed.
INTRODUCTION
- Authors should define IGD, listing its diagnostic criteria and distinguishing it from HG.
- Same consideration for PIU, authors cannot take for granted that readers know its definition.
- Authors should spend a few words describing the role of ANS.
- In the last paragraph of the introduction authors should state their objectives and aims, any RESULT (e.g., the number of subject included) must be avoided.
MATERIALS AND METHODS
- The number and characteristics of the included subjects should be listed in the RESULTS section. The results should only contain the required characteristics of the subjects (inclusion/exclusion criteria).
- Subjects should have been in common standardized conditions (e.g., no caffeine, no alcohol, nicotine intake at least in the 12 hours before the evaluation, no sleep deprivation, etc). Subjects should not have a previous diagnosis influencing ANS (e.g., diabetes) or taking any medication (e.g., B blockers). They should not have any previous diagnosis of psychiatric disorder other than (possibly) IGD. If all the above mentioned criteria could not have been met, they should be listed among the study limitations.
-
Author Response
The authors presented an interesting study illustrating the correlates of an ANS evaluation performed through HRV in HG. The manuscript is modern and will raise readers interest, but there are few minor concerns that may increase the overall quality if addressed.
Reply: Thank you for your helpful comments, we will provide detailed reply to these concerns below.
INTRODUCTION
- Authors should define IGD, listing its diagnostic criteria and distinguishing it from HG.
Reply: We added the definition and diagnostic criteria of IGD in the ‘Introduction’ section (Line 39).
- Same consideration for PIU, authors cannot take for granted that readers know its definition.
Reply: We added the definition of PIU in the ‘Introduction’ section (Line 50).
- Authors should spend a few words describing the role of ANS.
Reply: We added the role of ANS in the ‘Introduction’ section (Line 54).
- In the last paragraph of the introduction authors should state their objectives and aims, any RESULT (e.g., the number of subject included) must be avoided.
Reply: We added the objectives and aims in the last paragraph of the ‘Introduction’ section, and removed the ‘Result’.
MATERIALS AND METHODS
- The number and characteristics of the included subjects should be listed in the RESULTS section. The results should only contain the required characteristics of the subjects (inclusion/exclusion criteria).
Reply: We listed the number and characteristics of the included subjects in the ‘Results’ section (Line 211 and Table 1), and only required characteristics are contained.
- Subjects should have been in common standardized conditions (e.g., no caffeine, no alcohol, nicotine intake at least in the 12 hours before the evaluation, no sleep deprivation, etc). Subjects should not have a previous diagnosis influencing ANS (e.g., diabetes) or taking any medication (e.g., B blockers). They should not have any previous diagnosis of psychiatric disorder other than (possibly) IGD. If all the above mentioned criteria could not have been met, they should be listed among the study limitations.
Reply: We supplemented detailed inclusion criteria for the subjects in the ‘Materials and Methods’ section (Line 130), and all included subjects in this study were in common standardized conditions.

Round 2
Reviewer 1 Report
Comments and Suggestions for Authors
The authors have made all required corrections and clarifications to the text of the article.